# Autoantibody Formation and Mapping of Immunogenic Epitopes against Cold-Shock-Protein YB-1 in Cancer Patients and Healthy Controls

**DOI:** 10.3390/cancers12123507

**Published:** 2020-11-25

**Authors:** Ronnie Morgenroth, Charlotte Reichardt, Johannes Steffen, Stefan Busse, Ronald Frank, Harald Heidecke, Peter R. Mertens

**Affiliations:** 1Clinic of Nephrology and Hypertension, Diabetes and Endocrinology, Otto-von-Guericke University Magdeburg, Leipziger Str. 40, 39120 Magdeburg, Germany; ronnie.morgenroth@med.ovgu.de (R.M.); charlotte.reichardt@med.ovgu.de (C.R.); johannes.steffen@med.ovgu.de (J.S.); 2Clinic of Psychiatry and Psychotherapy, Otto-von-Guericke University Magdeburg, Leipziger Str. 40, 39120 Magdeburg, Germany; Stefan.busse@med.ovgu.de; 3AIMS Scientific Products GmbH, Galenusstr. 60, 13187 Berlin, Germany; ronald.frank@eu-openscreen.eu; 4CellTrend GmbH, im Biotechnologiepark 3, 14943 Luckenwalde, Germany; heidecke@celltrend.de

**Keywords:** cold shock proteins, autoantibody, peptide array, protein cleavage, antigenicity

## Abstract

**Simple Summary:**

Cold shock Y-box binding protein-1 plays a crucial role in cancerous cell transformation and proliferation. Experimental evidence links autoantibody formation with cancer diseases as well as YB 1 protein levels. Hence, we investigated autoantibody formation targeting YB-1 in cancer patients. Using recombinant proteins and specific peptide arrays, we mapped linear epitopes, which localize in the cold shock and C-terminal domain of the protein, in cancer patients that differ from healthy controls. Furthermore, cancer sera containing autoantibodies that target YB-1 extend the half-life of the YB-1 protein. Since extracellular YB-1 serves as a ligand for receptor Notch3 as well as TNFR1, this may contribute to aberrant signaling that promotes tumor development. In the clinical setting, we envision setting up detection assays for the immune response against YB-1, which may aid in screening for cancer.

**Abstract:**

Cold shock Y-box binding protein-1 participates in cancer cell transformation and mediates invasive cell growth. It is unknown whether an autoimmune response against cancerous human YB-1 with posttranslational protein modifications or processing develops. We performed a systematic analysis for autoantibody formation directed against conformational and linear epitopes within the protein. Full-length and truncated recombinant proteins from prokaryotic and eukaryotic cells were generated. Characterization revealed a pattern of spontaneous protein cleavage, predominantly with the prokaryotic protein. Autoantibodies against prokaryotic, but not eukaryotic full-length and cleaved human YB-1 protein fragments were detected in both, healthy volunteers and cancer patients. A mapping of immunogenic epitopes performed with truncated *E. coli*-derived GST-hYB-1 proteins yielded distinct residues in the protein N- and C-terminus. A peptide array with consecutive overlapping 15mers revealed six distinct antigenic regions in cancer patients, however to a lesser extent in healthy controls. Finally, a protein cleavage assay was set up with recombinant pro- and eukaryotic-derived tagged hYB-1 proteins. A distinct cleavage pattern developed, that is retarded by sera from cancer patients. Taken together, a specific autoimmune response against hYB-1 protein develops in cancer patients with autoantibodies targeting linear epitopes.

## 1. Introduction

Experimental and clinical evidence link cold shock protein overexpression with cancerous cell transformation [1]. For the prototypic member of this protein family, Y box-binding protein-1 (YB-1), intracellular functions have been described that link protein activities with cell proliferation, invasive cell growth [2], and metastasis formation [3]. Crucial regulatory events take place in the nucleus with changes of transcription of genes such as DNA polymerase α, PCNA, and EGF [4,5,6], as well as the cytoplasm with storage of mRNAs and orchestration of mRNA translation [7]. Beyond these, extracellular activities have been identified following the observation of active YB-1 protein secretion and binding to cell membrane-anchored receptors, such as Notch-3 [8,9]. The occurrence of extracellular YB-1 prompted additional studies to determine the applicability of YB-1 as a cancer protein marker in serum samples. Our group has performed extensive analyses of serum samples to determine the occurrence of extracellular YB-1 with newly generated monoclonal antibodies [10]. A small fragment with a relative molecular weight of 18 kDa was detected by Western-blotting in most serum samples from cancer patients that were absent in samples from healthy volunteers [10]. Notably, successful detection of the YB-1/p18 fragment by ELISA has not been achieved hitherto [10,11].

Extracellular YB-1 from non-transformed cells may have physiological functions on cell proliferation and inflammatory responses [12]. With cell transformation and YB-1 overexpression the immunogenicity of the protein likely changes. This phenomenon becomes evident when immunohistochemistry is performed with breast cancer tissue. In healthy tissue, most cells are negative for YB-1 protein with specific monoclonal antibodies. On the other hand, the same monoclonal antibodies clearly detected protein expression in most specimens with breast cancer [13]. Nuclear YB-1 detection goes along with a poor outcome of disease [14].

These findings prompted the current study to elucidate whether YB-1 derived from cancer cells exhibits different immunogenicity and may thus elicit an immunogenic response in cancer patients. Given the detection of YB-1 protein in serum samples, even in the absence of cell lysis, an immunogenic response may incite.

A review of the literature reveals that autoantibodies targeting YB-1 protein have been described in patients diagnosed with autoimmune diseases such as systemic sclerosis, primary biliary cholangitis autoimmune hepatitis (PBC-AIH) as well as autism spectrum disorders [15,16,17]. Jeoung et al. applied recombinant GST-YB-1 to determine autoantibody frequency in rheumatic diseases, with systemic sclerosis showing the highest prevalence (44%) followed by SLE (14%), whereas healthy controls mostly had no autoantibodies (7%). Braunschweig et al. investigated autoantibodies in mothers of children with an autism spectrum disorder, a neurodevelopmental disease. Autoantibodies targeting YB-1 protein were identified in 31% of affected patients whereas 23% of individuals in the control group were positive. In primary biliary cholangitis (PBC) and PBC-autoimmune hepatitis overlap syndrome (PBC-AIH-OS), both diseases of the hepatobiliary system, Nguyen et al. searched for autoantibodies to distinguish between both pathologies. They found YB-1 autoantibodies in both groups with a prevalence of about 30–35% [15,16,17].

The aforementioned studies tested for the prevalence of autoantibodies, however, did not analyze immunogenic epitopes, fragment-specific autoantibodies, and degradation characteristics. The dysregulation of naturally occurring autoantibodies may lead to specific autoimmune diseases and also cancer [18,19].

Here we report an in-depth study on the prevalence of autoantibodies directed against YB-1 protein in healthy volunteers versus cancer patients. Our experimental setup included different recombinant protein preparations from pro- and eukaryotic sources to detect autoantibodies. In addition, a peptide array with overlapping residues was designed that allowed to map linear epitopes recognized by autoantibodies. Lastly, a time-course experiment was performed to elucidate degradation patterns of spiked recombinant YB-1 protein in the presence and absence of autoantibodies within serum-samples.

## 2. Results

### 2.1. Recombinant YB-1 Protein Undergoes Spontaneous Cleavage

Our goal was to evaluate the occurrence of autoantibodies directed against cold shock Y-box binding protein-1 in cancer patients in comparison to healthy controls. To test for such an autoimmune response, heterologous expression systems were set up to express and purify recombinant human Y-box binding protein-1 in *E. coli* (hYB-1-*E. coli*) as well as immortalized human embryonic kidney cells HEK293 (hYB-1-HEK). The eukaryotic HEK293 expression system allows one to recapitulate posttranslational protein modifications, whereas these are not present in prokaryotic proteins derived from *E. coli*. Introduced vectors allowed to ectopically express proteins with hexahistidine- or GST-tags in the prokaryotic system and a Flag-tag in the eukaryotic proteins fused at the N-terminus (Figure 1A). The successful expression and affinity purification of these proteins through nickel and FlagM2 coated magnetic beads was ascertained by Western blotting with antibodies directed against the respective tags. Additionally, peptide-derived affinity-purified antibodies were generated against different epitopes localized within the protein N-terminus (aa 21–33), cold shock domain (85–96), and C-terminus (187–213 and 296–312) were applied (depicted in Figure 1A). The results confirmed the expression of full-length YB-1 protein, at the same time revealed a distinct pattern of cleaved protein fragments (Figure 1B). In the literature similar cleavage pattern for YB-1 protein had been reported and visualized before, however, has not been examined in detail. For instance [20,21] reported on the ectopic expression of YB-1 and showed Western blots with cleavage patterns of the recombinant proteins. With anti-His-tag antibody (Ab) several fragments ranging from 10 to 40 kDa are detected, that are denoted <2–10 in Figure 1B,C. With eukaryotic Flag-YB-1 there was only one additional band with full-length protein both at about 50 kDa. Notably, two very strong bands were present with His-tagged YB-1 at relative molecular size of 18–22 kDa that were detected with anti-N-term Ab only. One explanation may be that these are devoid of the His-tag at the terminal end and co-purify by the Ni-column affinity process. Corresponding bands with anti-His antibody was very weak (indicated by <6 and 7). Notably, there were also bands detected at 56, 58, and 70 kDa (indicated by * and **) with prokaryotic His-YB-1 protein that exceeds the reported relative molecular weight of YB-1 protein (50 kDa). Likely, these are aggregates of YB-1 protein derivatives co-purified by the expression and purification protocol. His-YB-1 and Flag-YB-1 protein cleavage patterns are complex (Figure 1C). Notably, the cleavage pattern markedly differed between *E. coli*- and HEK-derived YB-1 proteins with fewer cleavage fragments with the latter, although protease inhibitors and EDTA were added to the buffer and snap freezing in liquid nitrogen and storage at −80 °C was performed. Zhang et al. raised the possibility of an 11 amino acid extension of the C-terminal domain with eukaryotic YB-1, which stabilizes the protein and may prevent degradation. Those protein modifications are lacking in prokaryotic cold shock proteins [22,23]. Unexpectedly, bands corresponding to higher molecular weight than full-length YB-1 were detected as well. Possible explanations refer to protein homo-multimerization, enzymatic dimerization as well as CSD dimerization as reported before [24,25,26,27], or posttranslational modifications such as polyADPribosylation [28].

To obtain further information on the cleavage pattern another set of recombinant proteins similarly tagged at the N-terminus, however, with truncations at specific residues were tested. These GST-YB-1 protein constructs are depicted in Figure 2A, which allow for an even more precise mapping of putative cleavage sites. While the anti-GST-antibody can detect all recombinant protein derivatives, peptide-derived anti-YB-1-antibodies only detect their epitope containing GST-YB-1 fusion proteins, underlining their epitope specificity (Appendix A). The overall picture obtained with these constructs confirmed the spontaneous cleavage of the full-length protein at defined sites mapping to the residues provided in Figure 2C. This map on the “spontaneous” cleavage sites may also provide a hint at cleavage-related “neo-epitopes” that may be recognized by the immune system (see peptide array below). Another conclusion that may be drawn from the cleavage pattern seen in Figure 2B is that the processing of the YB-1 protein begins at the protein C-terminus and ends at the N-terminus, given that most protein fragments were detected with the antibodies directed at the protein N-terminus but not the C-terminus. This is a rather vague conclusion on the relative half-lives of the protein fragments, nevertheless, the processing must be dynamic even in the presence of protease inhibitors that were added to the buffer solution before dialysis of the recombinant proteins.

### 2.2. Autoantibody Pattern against Cold Shock Protein YB-1 Differs between Cancer Patients and Healthy Controls

To test for autoantibodies directed against cold shock YB-1 protein in healthy controls and cancer patients recombinant pro- and eukaryotic YB-1 proteins were separated by SDS-PA gel electrophoresis. Thereafter human serum samples were used as “primary” antibodies and mouse anti-human Fc-IgG antibodies followed by peroxidase-conjugated anti-mouse antibodies were applied to detect binding of serum IgG to the recombinant proteins. A complex pattern was seen with prokaryotic His-YB-1 protein that includes ~ 8 prominent bands (Figure 3A). Notably, strong bands corresponding to the size of full-length His-YB-1 protein were seen in 8 out of 15 cancer patients, but in only 3 out of 14 controls. The pattern of protein fragments appeared more complex with cancer patients, especially the one denoted p18 (#6) are more often present in cancer patients (~18 kDa). Similarly, high molecular weight bands >100 kDa are more often detected with cancer serum samples. The technique does not exclude the possibility of these bands corresponding to other proteins that co-elute in the recombinant protein purification process, however, a similar cleavage pattern of His-YB-1 was seen (Figure 3, lane 1). High molecular weight bands were also detected with recombinant protein alone and anti-His antibody, suggesting multimerization (indicated by *).

Taken together these results demonstrated a strong immunological response against *E. coli-*derived His-YB-1 that differed between healthy controls and cancer patients. A possible explanation for the missing detection of eukaryotic YB-1 protein by autoantibodies in serum samples could be the posttranslational YB-1 modifications or the lack of protein degradation, which could lead to conformational changes despite SDS-PA and heating.

### 2.3. YB-1 Phosphorylation Does Not Interfere with YB-1 Autoantibody Binding

As serum samples lacking for autoantibodies against eukaryotic Flag-YB-1 we raised the question if posttranslational modifications like phosphorylation are hindering autoantibody binding. In return, we set up a dephosphorylation assay using λ-phosphatase. Pervanadate stimulated and unstimulated Jurkat E6 T-cells were used for λ-phosphatase activity control. Both cells and eukaryotic Flag-YB-1 were incubated for 24 h at 30 °C with λ-phosphatase for dephosphorylation. Also, Flag-YB-1 was incubated overnight at 30 °C to exclude autoantibody binding due to spontaneous fragmentation or conformational changes during the incubation time. After blotting specific antibodies were applied.

As shown in Figure 4A (lane 1–4) λ-phosphatase successfully dephosphorylated the Jurkat T-cells. While untreated Flag-YB-1 decays over time in 2 additional N-terminal fragments at about 40 kDa and 35 kDa (4B, lane 3 >3 and >4) as well as corresponding C-terminal (4B, lane 5 >5 and >6) fragments, dephosphorylation leads to a delayed protein fragmentation (Figure 4B, lane 2, 4 and 6). This finding stands in line with previously reported results showing a destabilization of protein conformation with YB-1 phosphorylation [22] Protein band >2 at about 50 kDa most likely reflects untagged eukaryotic YB-1, while it is not detected by Flag-targeting antibody. In Figure 4C treated and untreated Flag-YB-1 were probed with serum from tumor patients (lane 3–6) and healthy controls (lane 7–10). Flag specific antibody was used as control (lane 1–2). All serum samples show unspecific reactions against lanes containing only sample buffer (Figure 4C, compare original blot). Nonetheless, YB-1 phosphorylation does not hinder autoantibody binding to eukaryotic Flag-YB-1 protein.

### 2.4. Epitope Mapping with Abbreviated YB-1 Protein Constructs and YB-1 Peptide Array

Next, we wished to apply the same technique of autoantibody detection with *E. coli-*derived GST-tagged YB-1 proteins that have already been introduced above to map antigenic epitopes. Advantages of these proteins are their predefined compositions and knowledge about cleavage patterns.

Results obtained with six representative serum samples from healthy controls and cancer patients each are shown in Figure 5A,B. With most serum samples autoantibodies against GST-YB-1 fusion proteins were visualized, however to a different extent in the two cohorts. With cancer serum samples most prominent bands were detected with constructs ∆2, ∆4, ∆8, and ∆9. These bands were markedly stronger than the ones seen in healthy controls (compare lanes 4, 6, 8, and 9)

A summary of the results obtained with all serum samples is depicted in Figure 5C, where band intensities are distinguished between weak and strong. The YB-1 protein is schematically drawn with subdomains defined by the GST constructs. The minimum binding regions identified with these constructs are highlighted. Notably, most patients exhibit autoantibodies recognizing domains residing in protein C- and N-terminus at the same time. This result may be explained by two distinct immunogenic regions. GST-tag protein alone was also blotted in these experiments. Sera with anti-GST activity were excluded from the analyses.

Next, experiments were designed to determine linear epitopes within the YB-1 protein that are recognized by the autoantibodies. A peptide array was set up that consisted of 79 individual peptides with 15 amino acids each. The peptides overlapped with 4 amino acids each and were chemically synthesized in parallel as spots on a cellulose membrane [29] (Appendix A). Serum samples were incubated with the peptide membrane as “primary” antibodies and binding of antibodies to the peptides was detected using a “secondary” anti-human IgG, IgA, and IgM alkaline phosphatase-coupled antibody. The array experiments were performed with serum samples from 13 healthy controls and 7 cancer patients with representative results shown in Appendix A. With healthy control serum, most arrays were blank or demonstrated only faint positive spots. On the contrary, there were several clusters of positive spots with serum from cancer patients, shown in the right panel (Appendix A). These may be grouped into five domains that are highlighted (Appendix A). A summary of all results is given in Figure 6A, where the intensity of the positive signals is graded into absent, weak, or strong. It becomes clear that strong autoantibody binding is predominantly seen with cancer patients. The corresponding amino acid sequences of peptide domains are shown in Figure 6B. These are found at the boundaries of the cold shock domain (aa 89–107), within the center of the cold shock domain, and at positively charged residues within the protein C-terminus. Notably, at serine 102, which is reported to be post-translationally modified [30], autoantibodies may bind.

Autoantibodies against GST-YB-1 cold shock domain fragment mostly lacking while are present in peptide array approach. Only a few serum samples elucidate such autoantibodies. This phenomenon may be due to YB-1 cold shock domain conformation. YB-1 CSD can refold after heating and underlies c [25,31]. Under consideration of accomplished peptide arrays, it seems more likely that ∆4 fragment is immunogenic by containing YB-1 cold shock domain. Considering densitometry readings CSD domain, especially peptide 22 and 23 as well as rear C-terminal peptide 70 are most immunogenic.

Autoantibody epitope pattern on the peptide array show specific differences between cancer patients and healthy controls. Linear epitopes recognized within the protein C-terminus are located at amino acid sequences 129–159, 181–211, 225–255, and 269–291 (Figure 6B). Notably, all linear epitope sequences can be linked to the cleavage sites of the YB-1 protein seen in our characterization of recombinant proteins (cp. Figure 1C and Figure 2C).

### 2.5. Endogenous YB-1 Serum Levels Do Not Differ between Healthy Controls and Cancer Patients; However, Addition of Recombinant Protein Reveals Retarded Degradation in Cancer Patients

In the following, we determined endogenous YB-1 protein levels in serum samples from healthy controls and cancer patients using two distinct peptide-derived polyclonal antibodies directed against different epitopes within the YB-1 protein (compare Figure 1A). These blots reveal that the overall YB-1 concentration is slightly higher in cancer patients (full-length protein detected at 50 kDa, lanes 5–10 (Appendix A). With anti-YB-1 N-terminal a smaller fragment with a relative molecular size of 22 kDa is detected, which is more abundant in the cancer patient’s serum samples. Notably, additional higher molecular weight bands indicated by “*” are detected with all antibodies, which are most prominently seen with anti-YB-1 C-terminal antibody, concordant to YB-1 C-terminal multimerization [24,25]. In cancer serum samples, the band corresponding to <*2 was more abundant than in healthy controls with anti-YB-1 Cold shock antibody. These findings indicate that the overall abundance of the YB-1 protein and the cancer serum samples is higher and that a processed N-terminal fragment is more abundant in these serum samples.

In the following, we wished to analyze how the presence of autoantibodies may affect the degradation of YB-1 protein and to this extent spiked the serum samples with recombinant pro- and eukaryotic YB-1 proteins, both of which harbor a unique tag that can be detected by Western blotting. The prokaryotic His-YB-1 and eukaryotic Flag-YB-1 protein were incubated at 37 °C for 30 min, 1 h, and 16 h with human serum samples collected from healthy volunteers and cancer patients (Figure 7A,B). The blots revealed a cleavage pattern of the added prokaryotic His-YB-1 protein with bands appearing at ~ 9, 18, 33, and 45 kDa (<*1–<*5), which did not differ between the cohorts (representative blots of 4 different serum samples are shown, each). In contrast, there was a marked difference in the banding pattern with eukaryotic Flag-YB-1 protein. Here, two additional novel bands appeared within 30 min incubation with serum sample in the control’s specimens (<#2–<#3), but not in the cancer specimens. In the latter, the banding appeared over time and were only seen at around 16 h (Figure 7B). Thus, the results suggest that the degradation of eukaryotic YB-1 protein is retarded. Notably, Figure 4B and Figure 7B pinpoint an acceleration YB-1 degradation in the absence of YB-1 autoantibodies, alternatively, healthy serum may contain more serum protease activity than patient samples. Given that, extracellular functions may be abolished.

## 3. Discussion

Changes in the overall protein concentrations, aberrant expression, altered protein structure, mutations, and posttranslational modifications may all result in the development of autoantibodies in cancer diseases, as first described in the 1950s. A defect in tolerance and inflammation, as well as cellular death mechanisms, may also lead to an altered humoral immune response, which may be detected up to 5 years before the development of clinical signs. Tumor-associated autoantibodies in clinical practice may be of use to stratify patients and their immunological responses [32], early-on the high prevalence of autoantibodies against proteins such as p53, c-myc, HER2, NY-ESO-1, CAGE, MUC1, and GBU4-5 [33,34]. Several reviews have covered this topic and provide ideas on the underlying mechanisms [35]. In recent meta-analyses, the authors conclude that their use should be confined to combination with other indicators due to a low sensitivity [36].

The presented data on autoantibody generation against cold shock YB-1 protein are the first hints at an autoimmune response in cancer patients against this evolutionarily conserved protein. The results do not come as a surprise, given the previous reports on upregulated YB-1 protein expression in numerous cancer entities and distinct posttranslational modifications in the protein, such as phosphorylation, acetylation, or ubiquitination [30,37,38,39,40,41]. What is confusing at first hand is the general finding of autoantibody formation against YB-1 protein in healthy controls as well, which however is only seen with recombinant protein from *E. coli*. Thus, autoimmunity against cold shock proteins seems to be a common phenomenon even in healthy individuals, which however changes in cancer patients. Under consideration of the age in the healthy control group, there might be the appearance of naturally occurring autoantibodies with increasing age [42]. These explanations would stand in line with a slightly higher autoantibody detection ratio compared to others [15]. Linear epitopes were mapped throughout the protein derived from a prokaryotic host as well as synthetically peptide array with most immunogenic regions in the cold shock and rear C-terminal domain. Thereby the charged residues within the protein C-terminus seem to be especially immunogenic. The posttranslational modifications like phosphorylation within the protein appear to not influence hindering autoantibody binding. Nevertheless, with the occurrence of autoantibodies, we detect a smaller fragment denoted p18/YB-1, and the overall serum content of YB-1 seems to be increased. Such a retarded degradation of protein in the presence of autoantibodies may be one effect of functional relevance while lacking YB-1 autoantibodies leads to accelerated disruption of the protein, the other may be linked to the protein-receptor interactions. Extracellular YB-1 is a non-canonical ligand for receptor Notch-3 and may activate intracellular signaling cascades [8]. Receptor Notch-3 is a notorious molecule for cancer propagation [43]. The latter interactions may be very complex, e.g., with steric hindrance following binding of the antibody, and have not been analyzed within this study. Given that extracellular YB-1 is highly mitogenic [9] such an effect may translate into cell proliferation cues. Unexpectedly, the detection of YB-1 levels in serum samples revealed only slight differences between cancer patients and healthy volunteers. Although several studies evaluated YB-1 protein function by using truncated YB-1 protein derivatives, only YB-1 protein fragment aa 1–219 has been described [44]. We identified 8 potential YB-1 fragments by using full length and truncated recombinant YB-1 protein derivatives and specific YB-1 antibodies. Kim et al. hypothesized a degradation via the 26S-proteasome, which truncates at so-called PEST (proline, glutamate, serine, and threonine) sequences. Such sequences can be found at aa 1–26, 26–52, 118–137, 170–185, 205–231, 264–279, and 304–324 [21]. Furthermore, Pu et al. calculated the antigenicity of YB-1 protein at similar peptide residues [45]. These suggestions would stand in line with our findings regarding YB-1 protein fragmentation. We used N-terminal tagged recombinant YB-1 protein derivatives which leads to C-terminal YB-1 fragmentation. Further studies using C-terminal tagged YB-1 protein should be performed to complete these findings. In our hands, such blots mostly do not detect further fragments that are likely to degrade rapidly. Notably, the eukaryotic YB-1 protein does not reveal such an extensive fragmentation pattern. This could be due to protein modifications like C-terminal amino acid extension which stabilize protein structure. Posttranslational modifications such as phosphorylation modify protein conformation [22]. It will be of interest to further analyze the identified linear antigenic epitopes for posttranslational modifications that may take place in YB-1 protein. In some domains, key phosphorylation sites, such as serine 102, have been found [30].

Of note is the immunogenic response against prokaryotic YB-1 preparations, whereas YB-1 from eukaryotic expression systems is not detected by immunoglobulins from cancer patient serum. The differences in protein antigenicity may reside in other posttranslational modifications besides phosphorylation (such as acetylation, glycosylation, methylation, and ubiquitination), as well as alterations in protein structure, which are already known for cold shock proteins to take place [46,47,48]. In this respect, the autoimmune response in cancer patients may be regarded as a fundamental misdirected immune response with consequences in protein degradation and key functions of serum YB-1 protein. Another possibility is that bacterial YB-1 forms different complexes with partner proteins/DNA/RNA molecules, which may change the overall structure and immunogenic response. In the clinical setting, our findings may be applied by setting up detection assays on the immune response against YB-1, which may aid screens for (preclinical) cancer. Furthermore, it will be of interest to follow over time the changes in antibody formation, e.g., with successful eradication of cancer tissue by surgery and/or (chemo-) therapy. Such studies need larger cohorts with samples collected over time and will provide insight into the benefit of such assays in patient care.

## 4. Materials and Methods

### 4.1. Subjects

The study was approved by the local ethics committee at the University Hospital Aachen and Magdeburg. Following providing written informed consent healthy volunteers and cancer patients were recruited (EK 107/05; 159/11). The 15 cancer patients had a mean age of 62.8 ± 10.6 years (8 females, 7 male), the 14 healthy controls a mean age of 75.0 ± 6.9 years (8 females, 6 male). A description of both cohorts, especially of the cancer entities and co-morbidities is provided in Appendix A.

### 4.2. Materials

#### 4.2.1. Expression and Purification of Prokaryotic and Eukaryotic YB-1 Proteins

The expression and purification protocol for prokaryotic His- and GST- proteins have been published before [49,50]. Vectors were kindly provided by K. Kohno.

Briefly, a pRSET vector containing an insert coding for a hexahistidine T7 epitope-YB-1 fusion protein was transferred in *E. coli* bacteria. The expression was induced with isopropyl-β-D-thiogalactoside. The expressed protein was released by several sonication and freezing cycles. The purification of recombinant YB-1 was performed with Ni+ affinity columns (Thermo Fisher Scientific, Rockford, IL, USA) and the protein were eluted with elution buffer (Imidazole and phosphate buffer).

For eukaryotic protein, a pcDNA3/Flag-YB-1 (Invitrogen Rockford, IL, USA) vector encoding for Flag-tagged YB-1 fusion protein was transfected into HEK-293-T cells by calcium-phosphate precipitation, as described elsewhere [51]. Recombinant YB-1 protein was purified with Flag-M2-Sepharose beads (Sigma-Aldrich, Darmstadt, Germany) according to the manufacturer’s instructions.

#### 4.2.2. YB-1 Autoantibody Detection in Serum and Plasma Samples from Healthy Volunteers and Cancer Patients

Serum samples were immediately prepared and stored at −80 °C until use. Plasma samples were diluted 1:1 with PBS before freezing at −80 °C. Recombinant His-YB-1 (2.5 µg), Flag-YB-1 (1.5 µg) and GST-YB-1 proteins (FL 0.5µg; ∆1 0.5 µg; ∆2 0.5 µg; ∆3 0.5 µg; ∆4 1 µg; ∆5 1 µg; ∆8 1 µg; ∆9 1.5 µg) were used as antigens. After electrophoresis, proteins were blotted onto nitrocellulose membranes and blocked with 2.5% dry milk in TBST for 1 h at room temperature. Blots were probed with human serum samples diluted 1:200 in TTBS and incubated overnight at 4 °C. Bound antibodies were detected with secondary anti-human-IgG antibody (Sigma-Aldrich; 1:5000) and visualized with peroxidase-conjugated anti-mouse IgG antibody (GE Healthcare, Buckinghamshire, UK; 1:2000) using the ECL-system (Thermo Fisher Scientific) on an Intas gel imaging system (Intas, Göttingen, Germany). Densitometry readings were performed with LabImage (Kapelan, Leipzig, Germany).

#### 4.2.3. YB-1 Dephosphorylation

Jurkat E6 T-cells were stimulated with an irreversible phosphatase inhibitor (pervanadate; H_2_O_2_ with orthovanadate) for 2 min at room temperature. Pervanadate prepared and unprepared cells as well as Flag-YB-1 protein were incubated for 24 h at 30 °C with λ-phosphatase (New England BioLabs, Frankfurt, Germany) for dephosphorylation. Flag-YB-1 without λ-phosphatase was also incubated at 30 °C for 24 h. About half a million cells and 1.5 µg of Flag-YB-1 were used for SDS-PAGE.

#### 4.2.4. Peptide Array

Human YB-1 protein sequence is represented by 79 overlapping 15mer peptide fragments, each peptide sequence overlaps the predecessor by 11 amino acid residues. Hence there is a shift of 4 amino acid residues between spots. Probing the array membrane was performed as described [52]. After use, peptide arrays were stripped of bound protein and reused for the next sample. Densitometry readings were performed with LabImage (Kapelan).

#### 4.2.5. YB-1 Immunoblotting

Human serum/plasma (0.1/0.2 μL) was separated on 10% SDS-PAGE gels, transferred to nitrocellulose, blocked with 5% BSA in TBST, and incubated overnight at 4 °C with primary polyclonal anti-YB-1 C-terminal antibody (Sigma-Aldrich; 1:1000), anti-YB-1 cold-shock-domain antibody (Eurogentec, Seraing, Belgium; 1:1000) or Anti-YB-1 (N-terminal) antibody in rabbit (Eurogentec; 1:1000). Peroxidase-conjugated secondary goat-anti-rabbit-antibody (Southern Biotech, Birmingham, AL, USA; 1:5000) and the ECL system (Thermo Fisher Scientific) were used for detection.

#### 4.2.6. YB-1 Cleavage Assay

Recombinant YB-1 proteins were incubated with 0.1 µl human serum for 30′, 1 h, and 16 h at 37 °C. Afterward separated by SDS-electrophoresis, blotted on nitrocellulose membrane, and incubated overnight at 4 °C with primary polyclonal anti-YB-1 antibody (Sigma-Aldrich; 1:1000) or monoclonal anti-Flag antibody (Flag M2, Sigma-Aldrich; 1:2000). Peroxidase-conjugated secondary goat-anti-rabbit-antibody (Southern Biotech; 1:5000) and the ECL system (Thermo Fisher Scientific) were used for detection.

## 5. Conclusions

Our findings evidence that a strong humoral response with autoantibody generation against YB-1 protein with cancer disease. Protein degradation and functions may be affected by this autoimmune response. These results may open opportunities for interventions and early cancer diagnosis.

## Figures and Tables

**Figure 1 cancers-12-03507-f001:**
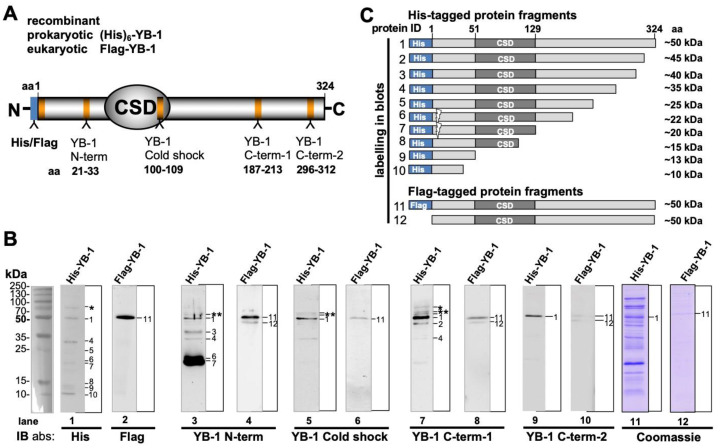
Recombinant YB-1 protein expression and spontaneous cleavage. (**A**) Schematic representation of recombinant His/Flag YB-1 protein with its N-terminus, C-terminus, and centrally localized cold shock domain. His- and Flag-tags are both located at the N-terminal ending. Locations of peptide epitope used for the immunization of rabbits to obtain polyclonal, affinity-purified antibodies are indicated. The nomenclature used is N-term, Cold shock, C-term-1, and C-term-2. (**B**) Immunoblotting and detection of recombinant His- (2.5 µg) and Flag (1.5 µg)-tagged YB-1 proteins with monoclonal anti-His and anti-Flag antibody, polyclonal anti-YB-1 antibodies, and Coomassie staining by 50 kDa. His-YB-1 presents several fragments reaching from 10 to 40 kDa. Flag-YB-1 presents only 1 additional band of 45 kDa relative molecular size. All antibodies are able to detect YB-1 full-length protein at the reported relative molecular weight of 52 kDa. Prokaryotic His-YB-1 degrades into several fragments (denoted 3 through 10), whereas eukaryotic Flag-YB-1 seems to be more stable and exhibits only one additional protein fragment at around 48 kDa (indicated by <2). There are two strong bands for His-YB-1 at 18–22 kDa detected with anti-N-term Ab only. Possibly these are co-purified protein fragments enriched by the Ni-column affinity process that is devoid of the His-tag at the terminal end. The corresponding band with anti-His antibody is very weak (indicated by >6 and >7). Notably, with prokaryotic His-YB-1, there are also bands detected at 56, 58, and 70 kDa (indicated by * and **), thus exceeding the reported relative molecular weight of YB-1 protein. Likely, these are aggregates of YB-1 protein derivatives that are not discernible by the SDS-PAGE procedure. (**C**) A schematic His-/Flag-YB-1 protein cleavage pattern was a set-up from the immunoblotting results. Full-length His-YB-1 (indicated by <1) may degrade into 9 additional fragments. Immunoblotting of Flag-YB-1 shows one additional fragment.

**Figure 2 cancers-12-03507-f002:**
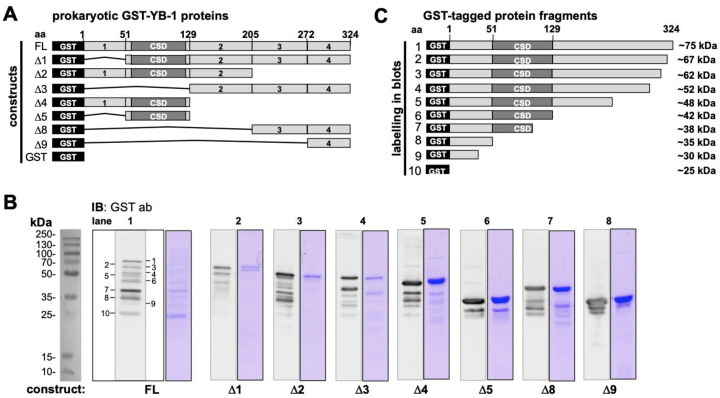
Characterization of spontaneous cleavage of recombinant prokaryotic YB-1 protein derivatives with a GST-tag in the protein N-terminus. (**A**). Scheme of recombinant GST-YB-1 fusion proteins with FL (0.5 µg) reflecting full-length YB-1. ∆1 (0.5 µg) lacks the N-terminal domains and presents the cold shock domain and C-terminal domains of YB-1. ∆2 (0.5 µg) extends from aa 1 to 205 and lacks part of the C-terminal alternative positive and negatively charged domains 3 and 4; ∆3 (0.5 µg) only consists of the charged C-terminal domains 2 through 4; ∆4 (1 µg) comprises the N-terminus and the CSD; ∆5 (1 µg) represents the cold shock domain; ∆8 (1 µG) includes aa 208–324; ∆9 (1.5 µg) reflects the rear C-terminal ending reaching from aa 272–324. (**B**) Immunoblotting of GST-YB-1 protein samples with anti-GST-antibody and Coomassie staining. GST-tag is of 25 kDa molecular size. Full-length GST-YB-1 has a relative molecular weight of 75 kDa. There are around 8 protein bands besides the GST-tag that are detected by the anti-GST antibody, which are denoted 1–5, 8, 9. (**C**) GST-YB-1 cleavage pattern based on immunoblotting.

**Figure 3 cancers-12-03507-f003:**
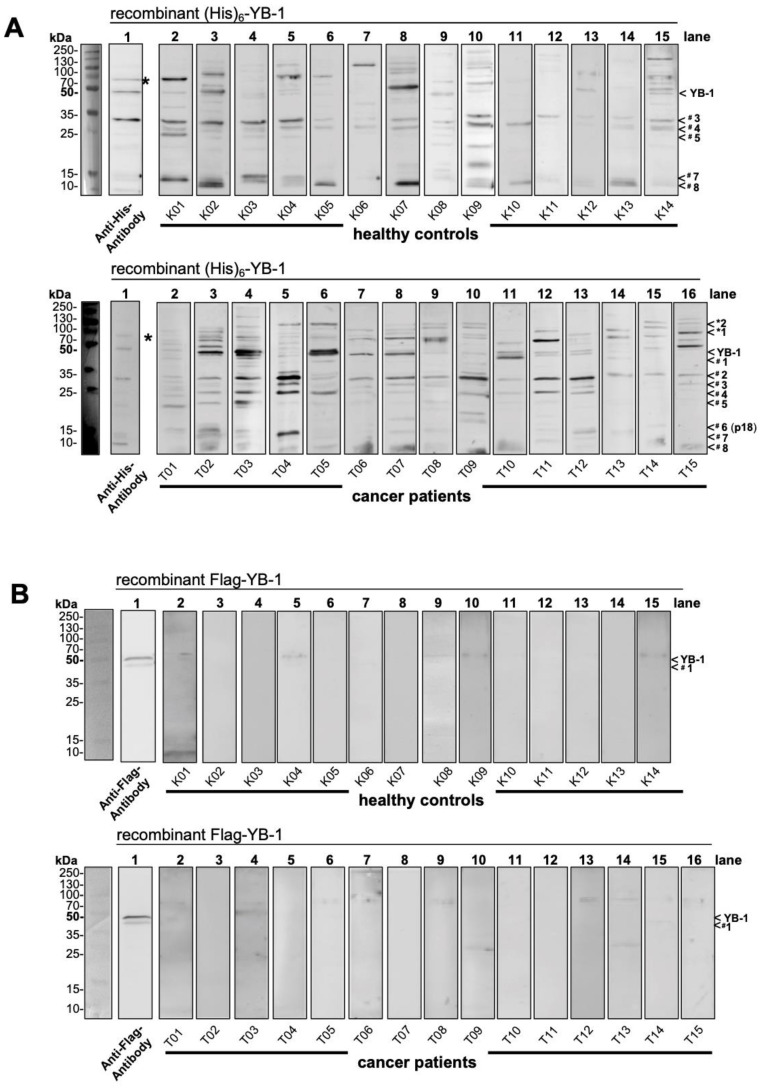
Autoantibody detection with recombinant YB-1 proteins. 2.5 µg of His-tagged and 1.5 µg of Flag-tagged YB-1 protein was separated by SDS-PAGE and serum samples from healthy controls and cancer patients were used to probe the blotted proteins as “primary” antibodies. Anti-human IgG Fc antibodies were added as the secondary antibody, followed by visualization with a tertiary anti-mouse antibody. (**A**) His tagged YB-1 protein is detected at about 50 kDa molecular weight by 1 control and 8 cancer patients. Both groups reveal protein bands from 10 kDa until 130 kDa molecular size. Cancer patients reveal antibodies against 8 fragments (#1–#8). Fragment denoted p18 (#6) seems specific in cancer. Bands of molecular size >100 kDa (*1, *2) are more common in cancer patients. (**B**) Autoantibodies against Flag-tagged YB-1 protein were absent in both cohorts. Only weak bands were seen, not corresponding to anti Flag antibody detected bands at about 50 kDa and 45 kDa (YB-1, #1). The same experimental approach was chosen for the recombinant Flag-tagged antibody expressed in HEK293 cells. Here, the recombinant protein did not degrade and was detected as a duplicate band ~50 kDa (Figure 3B, lane 1). Only faint bands were detected in some serum samples that however were of slightly higher relative molecular weight than full-length YB-1 (~54 kDa).

**Figure 4 cancers-12-03507-f004:**
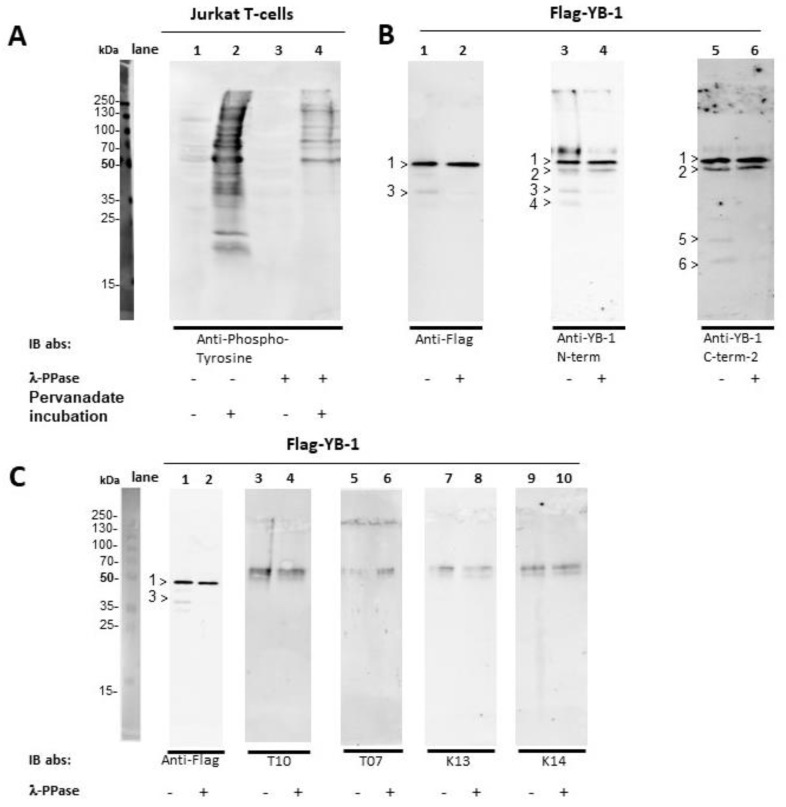
Dephosphorylation of Jurkat E6 T-cells and eukaryotic Flag-YB-1 Jurkat E6 T-cell extracts were prepared in the presence and absence of irreversible phosphatase inhibitor pervanadate. In a second step, the cell lysates, as well as the Flag-YB-1 protein, were incubated for 24 h at 30 °C with λ-phosphatase for dephosphorylation. Flag-YB-1 without the addition of λ-phosphatase was also incubated at 30 °C for 24 h. (**A**) About half a million cells were used for SDS-PAGE. Anti-phosphotyrosine antibody [4G10] was applied to Jurkat T-cells to demonstrate λ-phosphatase activity (lane 1–4). (**B**) To determine potential cleavage of Flag-YB-1 protein during incubation 1.5 µg of Flag-YB-1 was used for SDS-PAGE. Flag specific, as well as N-terminal and C-terminal targeting antibodies, were applied. Untreated Flag-YB-1 shows 2 N-terminal fragments at about 40 kDa and 35 kDa (lane 3, >3 and >4) with 2 corresponding C-terminal fragments (lane 5, >5 and >6). Fragment >2 (50 kDa) probably reflects untagged eukaryotic YB-1 as it is not detected by Flag specific antibodies. Dephosphorylated Flag-YB-1 presents a retarded fragmentation (compare lanes 1, 3, 5 with 2, 4, and 6). (**C**) 1.5 µg of untreated and dephosphorylated Flag-YB-1 protein were probed with a serum of tumor patients (lane 3–6) and healthy controls (lane 7–10). Flag specific antibody served as control (lane 1–2). All serum samples show unspecific reactions against lanes containing only sample buffer (compare original blots). None of the serum samples presents autoantibodies against eukaryotic Flag-YB-1.

**Figure 5 cancers-12-03507-f005:**
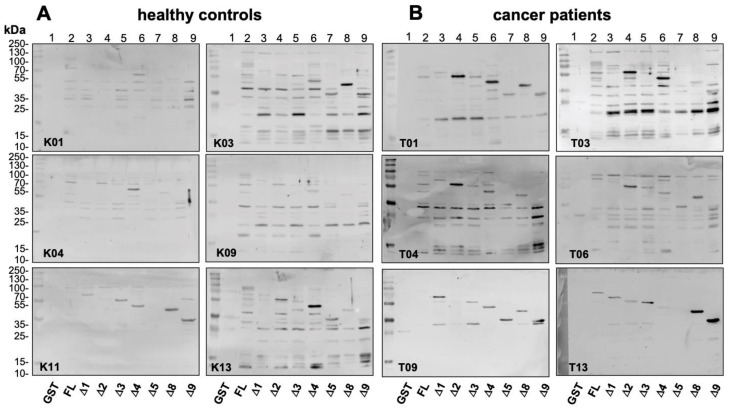
Epitope mapping with abbreviated YB-1 protein constructs. GST-tagged YB-1 proteins from Figure 2A were separated by SDS-PAGE and probed with serum samples from healthy controls and cancer patients as “primary” antibodies. Anti-human IgG Fc antibodies were added as the secondary antibody, followed by visualization with a tertiary anti-mouse antibody. (**A**,**B**) Six representative serum samples from the healthy controls and cancer patients are shown. In most samples’ autoantibodies against GST-YB-1 fusion proteins were visualized, however to a different extent in the two cohorts. In cancer patients, the band intensities seem stronger and the most prominent band detection is seen with constructs ∆2, ∆4, ∆8, and ∆9 (compare lanes 4, 6, 8, and 9 with other lanes). (**C**) Summary of all tested individuals. Controls (above) and cancer patients (below) are presented with their minimum autoantibody epitopes based on GST-YB-1 abbreviated protein construct blots. Signal of detected fragments are depicted in none (<10 M), weak (10–20 M), and strong (>20 M). For densitometry, readings compare with Appendix A.

**Figure 6 cancers-12-03507-f006:**
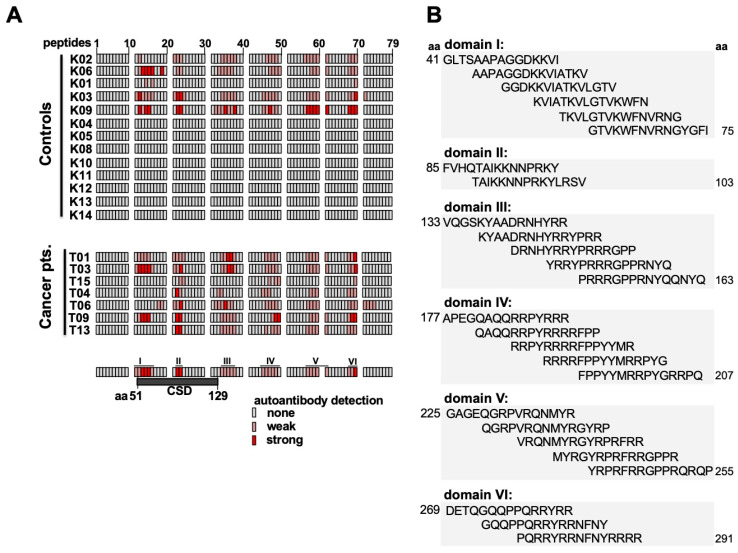
Mapping of autoantibodies against linear epitopes within cold shock protein YB-1 by overlapping peptide array. (**A**) Summary of all probed serum/plasma samples with detected peptides highlighted. Signal of detected fragments are semi objectively depicted none, weak (<100k), and strong (>100k). For densitometry readings compare with Appendix A. (**B**) The amino acid sequence of common cancer epitopes from Figure 6A.

**Figure 7 cancers-12-03507-f007:**
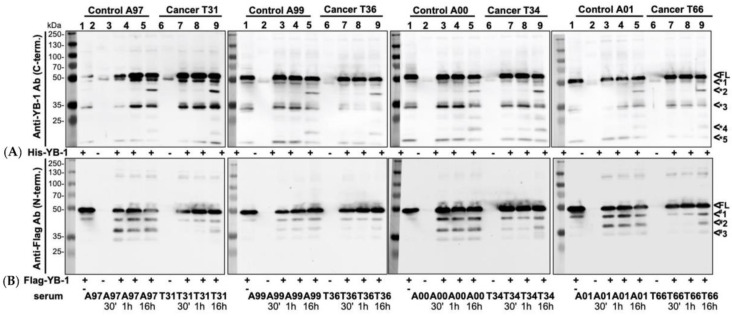
Effect of serum autoantibodies directed against cold shock protein YB-1 on protein degradation. (**A**) Prokaryotic His-YB-1 was incubated for 30 min, 1 h, and 16 h with a serum of healthy volunteers (lane 3–5) and cancer patients (lane 7–9) at 37 °C. His-YB-1, healthy and cancer serum alone were also applied (lane 1, 2, 6). His-YB-1 decays in 3 fragments of about 48 kDa, 32 kDa, and 15 kDa (*1, *3, *5) molecular weight. Serum probes show no fragmentation. Healthy and cancer incubated His-YB-1 show a stronger signal than His-YB-1 alone. After 16 h of incubation 2 additional fragments of approximately 40 kDa (*2) and 20 kDa (*4) molecular size were detected. (**B**) Eukaryotic Flag-YB-1 was incubated for 30 min, 1 h, and 16 h with a serum of healthy volunteers (lane 3–5) and cancer patients (lane 7–9) at 37 °C. Flag-YB-1, healthy and cancer serum alone were also applied (lane 1, 2, 6). Flag-YB-1 alone decays in one further fragment of about 45 kDa (#1). The control group presents already after 30 min of incubation 2 further fragments of about 35 kDa and 30 kDa (#2, #3) molecular size. The Cancer group only shows one additional fragment at approximately 35 kDa.

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
