# Peer review of "Autoantibody Formation and Mapping of Immunogenic Epitopes against Cold-Shock-Protein YB-1 in Cancer Patients and Healthy Controls"

_cancers, 2020, doi:10.3390/cancers12123507_

Round 1

Reviewer 1 Report

In the current manuscript by Morgenroth et al. authors studied formation of antibodies against YB-1 in cancer patients. Authors attempted to identify the epitope of YB-1 which is recognized by anti-YB-1 antibody present in human serum.

Under normal conditions YB-1 is presumable intra-cellular protein. However, extracellular YB-1 has been detected in patients with cancer and may cause auto-immune response. Thus, the topic of investigation is relevant to the field of cancer research.

The manuscript is very well written, and provides enough background about investigated question. However, I believe that it will strongly benefit from providing additional data, changing some data representation and improving the discussion.

Major points:

  1. In the Figure 1 authors analyze the cleavage pattern of YB-1 purified from E. Coli and HEK cells using western-blotting. I suggest to include in this figure the picture of gels stained with Coomassie in order to demonstrate the purity of recombinant proteins. I am afraid that contaminant proteins that are co-purified with YB-1 may be recognized by the antibodies used in this study and interfere with the interpretation of the data.
  2. Detection of high molecular weight bands that are larger than full-length YB-1 (> 50 kDA) are especially surprising because all pre-existing YB-1 multimers should be disrupted when proteins are running in SDS-containing gels. For example, see Guryanov et al, 2012. How authors make sure that these bands are specific?
  3. In the Figure 1 the difference in cleavage pattern between YB-1 purified from E. Coli and HEK cells is striking. How do authors exclude the possibility that lower molecular weight YB-1 fragments are not forming due to the premature termination of translation? The fact that most of these fragments are recognized by N-terminal antibody is also in agreement with idea. 
  4. In the Figure 3 authors compare the recognition of His-YB-1 and FLAG-YB-1 by the serum isolated from healthy volunteers and cancer patients. Authors should indicate what is the amount of each recombinant YB-1 was used in this experiment. Were the amounts of His-YB-1 and FLAG-YB-1 are the same? The details of experimental procedure should be added to the figure description and methods section. The same applies to peptide assay.In addition, I am not convinced that “the lack of protein degradation” could explain the absence of the signal when FLAG-YB-1 was used. In the Figure 3A full-length YB-1 is easily detected by the same antibodies indicating that degradation is not required for antibody detection.Is that possible that proteins that are purified from E. Coli are more reactive with antibodies present in human serum due to some specific bacterial modifications?
  5. In the Figure 4 authors used a variety of different GST-tagged YB-1 fragments in order to pin-point the exact conformational epitope that is recognized by anti-YB-1 antibody. This is an important experiment that resulted in the identification of epitopes in N- and C-terminal domains of YB-1. However, I am not sure that epitope conformation can be studied by this approach because all proteins should be mostly unfolded when migrated in SDS-PAGE gel and lose their 3D-fold. I think authors should either explain better what they mean by conformational epitopes and how their experimental set up allows to investigate it, or to change the text.
  6. In the Figure 5C authors identified additional epitopes that are recognized by antibodies present in the serum of cancer patients. However, these epitopes are different form the ones highlighted in Figure 4C. Notably, peptides derived from cold-shock domain are detectable in Figure 5, however this domain was not identified in Figure 4. Using two independent approaches can be a powerful strategy to identify immunogenic YB-1 epitope that is recognized by antibodies. I suggest to the authors to reconcile the results obtained from both experiments in order to identify the most immunogenic part of YB-1.­

Minor points:

  1. In Figure 1 B fragment #7 in detected with YB-1-C-term-1 antibodies. How is it possible if this fragment does not have C-terminal domain (Figure 1C)?
  2. In the figure 1 C authors should indicate the amino acid size of indicated YB-1 fragments.
  3. In the figure 2B numbers of corresponding YB-1 fragments should be indicated next to each western blot. ­­

If the authors will address the above-mentioned points I will recommend this manuscript for the publication in Cancers.  

Author Response

In the current manuscript by Morgenroth et al. authors studied formation of antibodies against YB-1 in cancer patients. Authors attempted to identify the epitope of YB-1 which is recognized by anti-YB-1 antibody present in human serum.

Under normal conditions YB-1 is presumable intra-cellular protein. However, extracellular YB-1 has been detected in patients with cancer and may cause auto-immune response. Thus, the topic of investigation is relevant to the field of cancer research.

The manuscript is very well written, and provides enough background about investigated question. However, I believe that it will strongly benefit from providing additional data, changing some data representation and improving the discussion.

Major points:

  1. In the Figure 1 authors analyze the cleavage pattern of YB-1 purified from E. Coli and HEK cells using western-blotting. I suggest including in this figure the picture of gels stained with Coomassie in order to demonstrate the purity of recombinant proteins. I am afraid that contaminant proteins that are co-purified with YB-1 may be recognized by the antibodies used in this study and interfere with the interpretation of the data.

Coomassie stainings of the purified proteins were performed to elucidate the purity. Both recombinant protein purifications show additional bands, other than the recombinant protein itself. Besides proteolytic fragments, which are shown in Figure 1,2 and Suppl. Figure 3, there is the possibility of co-purified proteins as well as YB-1 multimers [1-4].

  1. Detection of high molecular weight bands that are larger than full-length YB-1 (> 50 kDA) are especially surprising because all pre-existing YB-1 multimers should be disrupted when proteins are running in SDS-containing gels. For example, see Guryanov et al, 2012. How authors make sure that these bands are specific?

As described by Willis et al. YB-1 can form SDS and heat stable dimers mediated by protein transglutaminase [1]. Moreover, Guryanov et al. described YB-1 multimerisation up to 440 kDa molecular size [2]. Yang et al. also showed the possibility of stable dimerization viathe CSD [3].

  1. In the Figure 1 the difference in cleavage pattern between YB-1 purified from E. Coli and HEK cells is striking. How do authors exclude the possibility that lower molecular weight YB-1 fragments are not forming due to the premature termination of translation? The fact that most of these fragments are recognized by N-terminal antibody is also in agreement with idea. 

Defective ribosomal proteins (DRiPs) describe premature terminated or misfolded proteins. This phenomenon is a common feature of viral infected cells [10]. Also, eukaryotic cells have those defective proteins, which are often ubiquitinated for degradation [11].  In prokaryotes, like E. coli, the DRiPS were not described. The presene of C-terminal fragments favor proteolysis rather than defective translation.

With regards to our phosphatase assay, eukaryotic Flag-YB-1 was incubated at 30°C for 24h. After SDS-PAGE and protein blotting Flag-YB-1 was visualized by the same procedure as described in Figure 1. Notably, FLAG-YB-1 also presents a cleavage pattern with 2 additional N-terminal fragments at about 40 kDa and 35 kDa (Figure 4B lane 2 >3 and >4), with corresponding C-terminal fragments (Figure 4B lane 5 >5 and >6). While Flag-YB-1 does not exhibit any proteolytic fragments, spontaneous protein fragmentation seems probable, especially when dephosphorylated.

  1. In the Figure 3 authors compare the recognition of His-YB-1 and FLAG-YB-1 by the serum isolated from healthy volunteers and cancer patients. Authors should indicate what is the amount of each recombinant YB-1 was used in this experiment. Were the amounts of His-YB-1 and FLAG-YB-1 are the same? The details of experimental procedure should be added to the figure description and methods section. The same applies to peptide assay. In addition, I am not convinced that “the lack of protein degradation” could explain the absence of the signal when FLAG-YB-1 was used. In the Figure 3A full-length YB-1 is easily detected by the same antibodies indicating that degradation is not required for antibody detection. Is that possible that proteins that are purified from E. Coli are more reactive with antibodies present in human serum due to some specific bacterial modifications?

The idea that autoantibodies are reacting to bacterial modifications is unlikely, as the peptide array utilizes synthetic peptides, which react well with the autoantibodies.

The amount of His-YB-1 and Flag-YB-1 were 2.5 µg and 1.5 µg protein per lane, respectively. GST-YB-1 proteins (FL 0.5 µg; D1 0.5 µg; D2 0.5 µg; D 3 0.5 µg; D 4 1 µg; D 5 1 µg; D 8 1 µg; D 9 1.5 µg). Peptide array was set up synthetically as described by Beutling and Frank [12, 13].

  1. In the Figure 4 authors used a variety of different GST-tagged YB-1 fragments in order to pin-point the exact conformational epitope that is recognized by anti-YB-1 antibody. This is an important experiment that resulted in the identification of epitopes in N- and C-terminal domains of YB-1. However, I am not sure that epitope conformation can be studied by this approach because all proteins should be mostly unfolded when migrated in SDS-PAGE gel and lose their 3D-fold. I think authors should either explain better what they mean by conformational epitopes and how their experimental set up allows to investigate it, or to change the text.

The text has been changed.

  1. In the Figure 5C authors identified additional epitopes that are recognized by antibodies present in the serum of cancer patients. However, these epitopes are different form the ones highlighted in Figure 4C. Notably, peptides derived from cold-shock domain are detectable in Figure 5, however this domain was not identified in Figure 4. Using two independent approaches can be a powerful strategy to identify immunogenic YB-1 epitope that is recognized by antibodies. I suggest to the authors to reconcile the results obtained from both experiments in order to identify the most immunogenic part of YB-1. ­

Autoantibodies against GST-YB-1 cold shock domain fragment are mostly lacking while present in peptide array approach. Only a few serum samples elucidate such autoantibodies. This phenomenon may be due to YB-1 cold shock domain conformation as well as autoantibody affinity. YB-1 CSD has the ability of refolding after heating [2]. Under consideration of accomplished peptide arrays, it seems more likely that D4 fragment is immunogenic by containing YB-1 cold shock domain. Considering densitometry readings CSD domain, especially peptide 22 and 23 as well as the C-terminal peptide 70 are most immunogenic.

Minor points:

  1. In Figure 1 B fragment #7 in detected with YB-1-C-term-1 antibodies. How is it possible if this fragment does not have C-terminal domain (Figure 1C)?

This issue has been removed from the figure.

  1. In the figure 1 C authors should indicate the amino acid size of indicated YB-1 fragments.

We addressed your point in the figure.

  1. In the figure 2B numbers of corresponding YB-1 fragments should be indicated next to each western blot. ­­

We think this does not fit to our figure idea. We prepared an extra figure in the supplementary data, which address your point.

If the authors will address the above-mentioned points I will recommend this manuscript for the publication in Cancers.  

­­

Figure 4A-C. Dephosphorylation of Jurkat E6 T-cells and eukaryotic Flag-YB-1.    
Jurkat E6 T-cells were stimulated with an irreversible phosphatase inhibitor (pervanadate) for 2 min at room temperature. Untreated and treated cells as well as Flag-YB-1 protein were incubated for 24h at 30°C with l-phosphatase for dephosphorylation. Untreated Flag-YB-1 was also incubated at 30°C for 24h.  
A. About half a million cells were used for SDS-PAGE. Anti-phosphotyrosine antibody [4G10] was applied to Jurkat T-cells to demonstrate l-phosphatase activity (lane 1-4).         
B. To determine potential cleavage of Flag-YB-1 protein during incubation 1.5 µg of Flag-YB-1 was used for SDS-PAGE. Flag specific as well as N-terminal and C-terminal targeting antibody were applied. Untreated Flag-YB-1 shows 2 N-terminal fragments at about 40 kDa and 35 kDa (lane 3, >3 and >4) with 2 corresponding C-terminal fragments (lane 5, >5 and >6). Fragment >2 (50 kDa) probably reflects untagged eukaryotic YB-1 as its not detected by Flag specific antibody. Dephosphorylated Flag-YB-1 presents a retarded fragmentation (compare lanes 1, 3, 5 with 2, 4 and 6).
C.
1.5 µg of untreated and dephosphorylated Flag-YB-1 protein were probed with serum of tumor patients (lane 3-6) and healthy controls (lane 7-10). Flag specific antibody served as control (lane 1-2). All serum samples show unspecific reactions against the sample buffer (compare original blots). None of the serum samples presents autoantibodies against eukaryotic Flag-YB-1.

  1. Willis, W.L.; Hariharan, S.; David, J.J.; Strauch, A.R. Transglutaminase‐2 mediates calcium‐regulated crosslinking of the Y‐Box 1 (YB‐1) translation‐regulatory protein in TGFβ1‐activated myofibroblasts. Cell. Biochemie 2013, 114 (12), 2753-2769.
  2. Guryanov, S.G.; Filimonov, V.V.; Timchenko, A.A.; Melnik, B.S.; Kihara, H.; Kutyshenko, V.P.; Ovchinnikov, L.P.; Semisotnov, G.V. The major mRNP protein YB-1: Structural and association properties in solution. Biophys. Acta 2013, 1834, 559-567.
  3. Yang, X-J.; Zhu, H.; Mu, S-R.; Wei, W-J.; Yuan, X.; Wang, M.; Liu, Y.; Hui, J.; Huang, Y. Crystal structure of a Y-box binding protein 1 (YB-1)-RNA complex reveals key features and residues interacting with RNA. JBC 2019, doi: 10.1074/jbc.RA119.007545
  4. Okamoto, T.; Izumi, H.; Imamura, T.; Takano, H.; Ise, T.; Uchiumi, T.; Kuwano, M.; Kohno, K. Direct interaction of p53 with the Y-box binding protein, YB-1: a mechanism for regulation of human gene expression. Oncogene 2000, 19, 6194-6202.
  5. Zhang, J.; Fan, J-S.; Li, S.; Yang, Y.; Sun, P.; Zhu, O.; Wang, J.; Jiang, B.; Yang, D.; Liu, M. Structural basis of DNA binding to human YB-1 cold shock domain regulated by phosphorylation. Nucleic Acids Res. 2020, 48 (16), 9361-9371.
  6. Schindelin, H.; Jiang, W.; Inouye M.; Heinemann, U. Crystal structure of CspA, the major cold shock protein of Escherichia coli. Nati. Acad. Sci. USA. 1994, 91, 5119-5123.
  7. Kloks, C.P.A.M.; Tessari, M.; Vuister, G.W.; Hilbers, C.W. Cold Shock Domain of the Human Y-Box Protein YB-1. Backbone Dynamics and Equilibrium between the Native State and a Partially Unfolded State. Biochemistry 2004, 43, 10237-10246.
  8. Mordovkina, D.; Lyabin, D.N.; Smolin, E.A.; Sogorina, E.M.; Ovchinnikov, L.P.; Eliseeva, I. Y-Box Binding Proteins in mRNP Assembly, Translation, and Stability Control. Biomolecules 2020, 10 (4), 591.
  9. Kljashtorny, V.; Nikonov, S.; Ovchinnikov, L.P.; Lyabin, D.; Vodovar, N.; Curmi, P.; Manivet, P. The Cold Shock Domain of YB-1 Segregates RNA from DNA by Non-Bonded Interactions. PLoS One 2015, doi.org/10.1371/journal.pone.0130318.
  10. Yewdell, J.W.; Antón, L.C.; Bennink, J.R. Defective ribosomal products (DRiPs): a major source of antigenic peptides for MHC class I molecules? Immunol.1996, 157 (5), 1823-1826.
  11. Mediani, L.; Guillén-Boixet, J.; Vinet, J.; Franzmann, T.M.; Bigi, I.; Mateju, D.; Carrà, A.D.; Morelli, F.F.; Tiago, T.; Poser, I.; Alberti, S.; Carra, S. Defective ribosomal products challenge nuclear function by impairing nuclear condensate dynamics and immobilizing ubiquitin. EMBO J 2019, 38 (15), doi: 10.15252/embj.2018101341.
  12. Frank, R. Spot-Synthesis: An easy technique for the positionally addressable, parallel chemical synthesis on a membrane support. Tetrahedron 1992, 48, 9217-9232.
  13. Beutling, U.; Frank, R. Epitope analysis using synthetic peptide repertoires prepared by SPOT synthesis technology. (2010) In: Antibody Engineering; Kontermann, R., S. Dübel, S.; Springer-Verlag: Berlin Heidelberg, Germany, 2010; Volume 1, pp. 537-572.

Reviewer 2 Report

In Morgenroth et al. the authors present data demonstrating that sera from cancer patients and healthy donors contain autoantibodies against YB1, an important regulator of transcription and translation in human cells. As the authors point out, previous work has also demonstrated that autoantibodies are raised against YB1 in certain disease states, such as systemic sclerosis.

The authors also demonstrate that YB1 is subject to proteolytic cleavage, another area that has been extensively covered with regards to YB1.

Overall, the data presented is of very high quality and the experiments are conducted with great care. My concern is that this data is mostly descriptive. There is no clear evidence of the importance or relevance of these antibodies. This is particularly true as they are present in both healthy donors and cancer patients. Further, the fact that the antibodies seem to only recognize bacterially produced YB1 brings further questions about the relevance.

Before publication, the authors must clarify the importance or relevance of this data in this instance. This is particularly true since autoantibodies against YB1 have already been shown to be relevant in other conditions.  

Experimental Points:

  1. The authors should provide Coomassie stained gels of their protein purifications. As they point out, it is possible that their antibodies are recognizing some co-purifying proteins. The purity of such preparations should be presented. How closely do the fragments identified by blotting line up with the purified proteins? That is, is increased intensity by blotting due to increased abundance of that proteolytic fragments?
  2. The authors suggest that post-translational modifications could be a contributing factor to the lack of immunogenicity of eukaryotically produced YB1. Indeed, YB1 is heavily post-translationally modified. Most of these modifications are phosphorylation events, although some acetylation and methylation has been reported. To test their hypothesis, the authors should attempt to dephosphorylate 293T derived YB1 with lambda protein phosphatase. If phosphates are preventing recognition, this should make the purified protein recognizable by the autoantibodies.

Minor Point:

  1. During formatting, it appears as if the “delta” symbol was changed to a spiral. This should be fixed.
  2. There are red lines underneath “domain” in figure 5.

Author Response

In Morgenroth et al. the authors present data demonstrating that sera from cancer patients and healthy donors contain autoantibodies against YB1, an important regulator of transcription and translation in human cells. As the authors point out, previous work has also demonstrated that autoantibodies are raised against YB1 in certain disease states, such as systemic sclerosis.

The authors also demonstrate that YB1 is subject to proteolytic cleavage, another area that has been extensively covered with regards to YB1.

Overall, the data presented is of very high quality and the experiments are conducted with great care. My concern is that this data is mostly descriptive. There is no clear evidence of the importance or relevance of these antibodies. This is particularly true as they are present in both healthy donors and cancer patients. Further, the fact that the antibodies seem to only recognize bacterially produced YB1 brings further questions about the relevance.

Before publication, the authors must clarify the importance or relevance of this data in this instance. This is particularly true since autoantibodies against YB1 have already been shown to be relevant in other conditions.  

Experimental Points:

  1. The authors should provide Coomassie stained gels of their protein purifications. As they point out, it is possible that their antibodies are recognizing some co-purifying proteins. The purity of such preparations should be presented. How closely do the fragments identified by blotting line up with the purified proteins? That is, is increased intensity by blotting due to increased abundance of that proteolytic fragments?

Coomassie staining of purified proteins were performed to elucidate protein purity. Both recombinant prokaryotic protein purifications show more bands than the recombinant protein itself. Besides proteolytic fragments which are shown in Figure 1,2 and Suppl. Figure 3 there is the possibility of co-purified proteins as well as YB-1 multimers [1-4].

Protein bands of Coomassie staining exceed YB-1 related protein bands detected by specific YB-1 antibodies. Most prominent protein band at about 22 kDa also detected with YB-1 N-term antibody shows most abundance, but do not correlate with high intensity when compared with human serum samples (compare Figure 3). Also, GST-YB-1 fragment D5 containing YB-1 cold shock domain has the same amount of protein as D4. Given that, YB-1 cold shock domain could be folded in a manner, which is not recognized by autoantibodies.

  1. The authors suggest that post-translational modifications could be a contributing factor to the lack of immunogenicity of eukaryotically produced YB1. Indeed, YB1 is heavily post-translationally modified. Most of these modifications are phosphorylation events, although some acetylation and methylation has been reported. To test their hypothesis, the authors should attempt to dephosphorylate 293T derived YB1 with lambda protein phosphatase. If phosphates are preventing recognition, this should make the purified protein recognizable by the autoantibodies.

As mentioned YB-1 can be heavily posttranslational modified [40-44]. We performed a dephosphorylation assay of eukaryotic Flag-YB-1 to elucidate immunogenic properties of YB-1 protein phosphorylation status. Unstimulated and pervanadate treated Jurkat E6 T-cells served as the positive control for λ-phosphatase activity. Phosphatase was incubated overnight with the cell lysates and Flag-YB-1 protein at 30°C. Also, Flag-YB-1 was treated overnight at 30°C without phosphatase and used as control. SDS-PAGE with about half a million cells and 1.5 µg of Flag-YB-1 was performed. After blotting specific antibodies shown in Figure 4A were applied overnight at 4°C. λ-phosphatase activity could be elucidated after incubation with anti-phosphotyrosine [4G10] and visualized with HRP conjugated secondary antibody and ECL (Figure 4A lane 1-4). Furthermore, Flag-YB-1 demonstrates spontaneous degradation with two additional N-terminal fragments with about 40 kDa and 35 kDa molecular size (Figure 4B lane 3  >3, >4). C-terminal antibody shows corresponding bands (Figure 4B lane 5 >5 and >6). Protein band at about 50 kDa (>2) most likely reflects untagged eukaryotic YB-1 while it´s not detected by Flag specific antibody. Notably, treatment with l-phosphatase delays YB-1 fragmentation (compare 4B lane 1 and 3 with 2 and 4). This finding stands in line with previous reported results showing a destabilizing protein conformation shift from folded to unfolded forms with YB-1 phosphorylation [5]. Also, as provided before, C-terminal domain of YB-1 plays a crucial role in stabilizing protein structure by electrostatic interactions [2, 5].
Untreated and dephosphorylated Flag-YB-1 were also probed with serum of tumor patients and healthy controls. All serum samples show unspecific reactions against lanes containing only sample buffer (Figure 4C). Nonetheless, YB-1 phosphorylation does not hinder autoantibody binding to eukaryotic Flag-YB-1 protein. Notably, Figure 4B and 7B pinpoint an acceleration YB-1 degradation in the absence of YB-1 autoantibodies, alternatively healthy serum may contain more serum protease activity than patient samples.

Minor Point:

  1. During formatting, it appears as if the “delta” symbol was changed to a spiral. This should be fixed.

This issue has been fixed.

  1. There are red lines underneath “domain” in figure 5.

This issue has been fixed.

­­

Figure 4A-C. Dephosphorylation of Jurkat E6 T-cells and eukaryotic Flag-YB-1.    
Jurkat E6 T-cells were stimulated with an irreversible phosphatase inhibitor (pervanadate) for 2 min at room temperature. Untreated and treated cells as well as Flag-YB-1 protein were incubated for 24h at 30°C with l-phosphatase for dephosphorylation. Untreated Flag-YB-1 was also incubated at 30°C for 24h.  
A. About half a million cells were used for SDS-PAGE. Anti-phosphotyrosine antibody [4G10] was applied to Jurkat T-cells to demonstrate l-phosphatase activity (lane 1-4).         
B. To determine potential cleavage of Flag-YB-1 protein during incubation 1.5 µg of Flag-YB-1 was used for SDS-PAGE. Flag specific as well as N-terminal and C-terminal targeting antibody were applied. Untreated Flag-YB-1 shows 2 N-terminal fragments at about 40 kDa and 35 kDa (lane 3, >3 and >4) with 2 corresponding C-terminal fragments (lane 5, >5 and >6). Fragment >2 (50 kDa) probably reflects untagged eukaryotic YB-1 as its not detected by Flag specific antibody. Dephosphorylated Flag-YB-1 presents a retarded fragmentation (compare lanes 1, 3, 5 with 2, 4 and 6).
C.
1.5 µg of untreated and dephosphorylated Flag-YB-1 protein were probed with serum of tumor patients (lane 3-6) and healthy controls (lane 7-10). Flag specific antibody served as control (lane 1-2). All serum samples show unspecific reactions against the sample buffer (compare original blots). None of the serum samples presents autoantibodies against eukaryotic Flag-YB-1.

Submission Date

30 August 2020

Date of this review

19 Sep 2020 01:35:42

  1. Willis, W.L.; Hariharan, S.; David, J.J.; Strauch, A.R. Transglutaminase‐2 mediates calcium‐regulated crosslinking of the Y‐Box 1 (YB‐1) translation‐regulatory protein in TGFβ1‐activated myofibroblasts. Cell. Biochemie 2013, 114 (12), 2753-2769.
  2. Guryanov, S.G.; Filimonov, V.V.; Timchenko, A.A.; Melnik, B.S.; Kihara, H.; Kutyshenko, V.P.; Ovchinnikov, L.P.; Semisotnov, G.V. The major mRNP protein YB-1: Structural and association properties in solution. Biophys. Acta 2013, 1834, 559-567.
  3. Yang, X-J.; Zhu, H.; Mu, S-R.; Wei, W-J.; Yuan, X.; Wang, M.; Liu, Y.; Hui, J.; Huang, Y. Crystal structure of a Y-box binding protein 1 (YB-1)-RNA complex reveals key features and residues interacting with RNA. JBC 2019, doi: 10.1074/jbc.RA119.007545
  4. Okamoto, T.; Izumi, H.; Imamura, T.; Takano, H.; Ise, T.; Uchiumi, T.; Kuwano, M.; Kohno, K. Direct interaction of p53 with the Y-box binding protein, YB-1: a mechanism for regulation of human gene expression. Oncogene 2000, 19, 6194-6202.
  5. Zhang, J.; Fan, J-S.; Li, S.; Yang, Y.; Sun, P.; Zhu, O.; Wang, J.; Jiang, B.; Yang, D.; Liu, M. Structural basis of DNA binding to human YB-1 cold shock domain regulated by phosphorylation. Nucleic Acids Res. 2020, 48 (16), 9361-9371.
  6. Schindelin, H.; Jiang, W.; Inouye M.; Heinemann, U. Crystal structure of CspA, the major cold shock protein of Escherichia coli. Nati. Acad. Sci. USA. 1994, 91, 5119-5123.
  7. Kloks, C.P.A.M.; Tessari, M.; Vuister, G.W.; Hilbers, C.W. Cold Shock Domain of the Human Y-Box Protein YB-1. Backbone Dynamics and Equilibrium between the Native State and a Partially Unfolded State. Biochemistry 2004, 43, 10237-10246.
  8. Mordovkina, D.; Lyabin, D.N.; Smolin, E.A.; Sogorina, E.M.; Ovchinnikov, L.P.; Eliseeva, I. Y-Box Binding Proteins in mRNP Assembly, Translation, and Stability Control. Biomolecules 2020, 10 (4), 591.
  9. Kljashtorny, V.; Nikonov, S.; Ovchinnikov, L.P.; Lyabin, D.; Vodovar, N.; Curmi, P.; Manivet, P. The Cold Shock Domain of YB-1 Segregates RNA from DNA by Non-Bonded Interactions. PLoS One 2015, doi.org/10.1371/journal.pone.0130318.
  10. Yewdell, J.W.; Antón, L.C.; Bennink, J.R. Defective ribosomal products (DRiPs): a major source of antigenic peptides for MHC class I molecules? Immunol.1996, 157 (5), 1823-1826.
  11. Mediani, L.; Guillén-Boixet, J.; Vinet, J.; Franzmann, T.M.; Bigi, I.; Mateju, D.; Carrà, A.D.; Morelli, F.F.; Tiago, T.; Poser, I.; Alberti, S.; Carra, S. Defective ribosomal products challenge nuclear function by impairing nuclear condensate dynamics and immobilizing ubiquitin. EMBO J 2019, 38 (15), doi: 10.15252/embj.2018101341.
  12. Frank, R. Spot-Synthesis: An easy technique for the positionally addressable, parallel chemical synthesis on a membrane support. Tetrahedron 1992, 48, 9217-9232.
  13. Beutling, U.; Frank, R. Epitope analysis using synthetic peptide repertoires prepared by SPOT synthesis technology. (2010) In: Antibody Engineering; Kontermann, R., S. Dübel, S.; Springer-Verlag: Berlin Heidelberg, Germany, 2010; Volume 1, pp. 537-572.

Round 2

Reviewer 1 Report

The manuscript has been significantly improved and can be accepted for the publication in Cancers. 

Author Response

Dear Editor,

thank you for your advices.

Kind regards,

Ronnie Morgenroth